

# Distinct patterns in the gut microbiota after surgical or medical therapy in obese patients

Daniel A. Medina[1], Juan P. Pedreros[1], Dannae Turiel[2], Nicolas Quezada[2], Fernando Pimentel[2], Alex Escalona[3] and Daniel Garrido[1]

[1] Department of Chemical and Bioprocess Engineering, Pontificia Universidad Católica de Chile, Santiago, Chile
[2] Department of Digestive Surgery, School of Medicine, Pontificia Universidad Católica de Chile, Santiago, Chile
[3] Department of Surgery, Faculty of Medicine, Universidad de Los Andes, Santiago, Chile

## ABSTRACT

Bariatric surgery is highly successful in improving health compared to conventional dietary treatments. It has been suggested that the gut microbiota is a relevant factor in weight loss after bariatric surgery. Considering that bariatric procedures cause different rearrangements of the digestive tract, they probably have different effects on the gut microbiota. In this study, we compared the impact of medical treatment, sleeve gastrectomy and Roux-en-Y gastric bypass on the gut microbiota from obese subjects. Anthropometric and clinical parameters were registered before, 6 and 12 months after treatment. Fecal samples were collected and microbiota composition was studied before and six months post treatment using 16S rRNA gene sequencing and qPCR. In comparison to dietary treatment, changes in intestinal microbiota were more pronounced in patients subjected to surgery, observing a bloom in *Proteobacteria*. Interestingly, *Bacteroidetes* abundance was largely different after six months of each surgical procedure. Furthermore, changes in weight and BMI, or glucose metabolism, correlated positively with changes in these two phyla in these surgical procedures. These results indicate that distinct surgical procedures alter the gut microbiota differently, and changes in gut microbiota might contribute to health improvement. This study contributes to our understanding of the impact of weight loss surgery on the gut microbiota, and could be used to replicate this effect using targeted therapies.

Corresponding author
Daniel Garrido, dgarridoc@ing.puc.cl

## INTRODUCTION

Obesity is a worldwide health problem that negatively affects quality of life. According to the *World Health Organization (2016)*, more than 1,900 million people over 18 years old have a body mass index (BMI) of 25 kg/m$^2$ or greater, and 600 million are catalogued as obese, with BMI $\geq$ 30 kg/m$^2$. Type 2 diabetes, cardiovascular disorders, certain cancers and asthma are comorbidities that show an increased risk in subjects with obesity.

The first line of treatment for obesity is medical treatment, which combines diet and physical activity. Unfortunately, the effectiveness of this approach appears to be only short term, since weight regain is common and not all patients respond similarly (*Kral et al., 2012*). In subjects with obesity and comorbidities, surgical procedures have been successful in controlling weight in the long term and reducing the incidence of related comorbidities, such as hypertension and type 2 diabetes (*Sjöström et al., 2007*; *Sjöström, 2008*; *Eldar et al., 2011*). These procedures are collectively known as bariatric surgery (BS). Indications for BS include a BMI more than 40 or a BMI more than 35 with medical comorbidities (*Mechanick et al., 2013*). BS can either restrict food intake (restrictive), or reduce nutrient absorption (malabsorptive) (*Buchwald et al., 2004*). Sleeve gastrectomy (SG) is an example of a restrictive procedure. It removes a significant portion of the stomach, decreasing its volume and leading to a significant reduction in the amount of food consumed (*Gumbs et al., 2007*). Meanwhile, Roux-en-Y gastric bypass (RYGB) is both restrictive and malabsorptive, creating a small stomach pouch connected to the proximal jejunum, reducing stomach volume to restrict food intake and bypassing food to the small intestine (*Tice et al., 2008*). Both procedures cause anatomical rearrangements that directly change gastrointestinal anatomy and function, accelerating food transit and altering hormonal regulation (*Tice et al., 2008*; *Tran et al., 2016*). While weight loss could be more pronounced in patients undergoing RYGB compared to SG after two years, the risk for post-surgical complications is greater in patients who have undergone RYGB (*Lager et al., 2016*). Other studies indicate that RYGB significantly outperforms SG in achieving glycated haemoglobin (HbA1C) values under 7.0% without medications (*Schauer et al., 2014*). In aggregate, these observations make interesting to understand the changes in the gut microbiota associated to both surgeries.

The intestinal microbiota has been shown to have a strong impact on host health and is considered a metabolic organ. It consists of a dense community of microorganisms that matches the number of cells of the human body (*Sender, Fuchs & Milo, 2016*). The influence of the gut microbiota is better exemplified at the metabolic level, since the microbiota synthesizes vitamins and amino acids absorbed by the epithelium (*LeBlanc et al., 2013*). Additionally, it is capable of fermenting complex dietary polysaccharides and other dietary sources, resulting in the production of short chain fatty acids (SCFA) such as acetate, propionate and butyrate (*Cook & Sellin, 1998*; *Hijova & Chmelarova, 2007*; *Morrison et al., 2016*). These acids modulate physiological processes in several tissues, such as insulin sensitivity, liver function and cholesterol metabolism (*Todesco et al., 1991*; *Demigné et al., 1995*; *Fushimi et al., 2006*; *Gao et al., 2009*; *Den Besten et al., 2013*). Furthermore, the gut microbiota plays important roles in the development of the immune system and the maintenance of intestinal epithelium integrity (*Sekirov et al., 2010*).

Certain studies have linked obesity with changes in the composition and metabolic function of the gut microbiota (*Bäckhed et al., 2004*; *Ley et al., 2005*; *Turnbaugh et al., 2006*; *Tremaroli et al., 2015*; *Palleja et al., 2016*). The gut microbiota is dominated by species that belong mainly to the *Firmicutes* and *Bacteroidetes* phylum, and to a lesser degree to *Actinobacteria*, *Proteobacteria* and *Verrucomicrobia* (*Qin et al., 2010*). In obese subjects, there has been an observed decrease in the relative proportion of the *Bacteroidetes/Firmicutes*

ratio, compared to lean people. Interestingly, this phenotype is transmissible to mice. Moreover, this proportion appears increased after weight loss on two low-calorie diets (*Ley et al., 2006*). These taxonomical differences in the gut microbiota of obese subjects might contribute to obesity in several ways, including energy extraction from the diet (*Turnbaugh et al., 2006*), mainly from SCFA, together with an increase in low-grade inflammation and altered bile acid metabolism (*Khan et al., 2016*).

Interestingly, bariatric surgery also induces important changes in the composition of the gut microbiota of patients undergoing these procedures (*Zhang et al., 2009*; *Li et al., 2011*; *Kong et al., 2013*). The main changes reported after surgical intervention include increases in *Proteobacteria* (*E. coli*, *Enterobacter spp.*), decreases in *Clostridium* and changes in *Bacteroides* and *Prevotella* (*Zhang et al., 2009*; *Furet et al., 2010*; *Li et al., 2011*; *Huttenhower et al., 2012*; *Kong et al., 2013*). Furthermore, it has been reported that these taxonomical and functional changes in the microbiota are stable nine years after RYGB intervention (*Tremaroli et al., 2015*).

Whether the observed changes in microbiota composition contribute to weight loss or whether they are just a consequence of the surgical procedure is unclear. In a mouse model, transfer of the gut microbiota from RYGB-operated to germ-free mice induced weight loss and decreased fat mass in comparison with germ-free animals colonized with microbiota from sham-operated animals (*Liou et al., 2013*). This suggested that the gut microbiota is an active player in weight loss in obesity surgery, and that weight loss is a transmissible trait of the microbiota post-surgery. On the other hand, a recent report showed that in one person, fecal microbiota from a healthy but overweight donor induced obesity (*Alang & Kelly, 2015*).

Sleeve gastrectomy and gastric bypass are common bariatric procedures that exert different physiological changes in the gastrointestinal tract, possibly inducing different changes in the gut microbiota that may contribute to different health outcomes. Unfortunately, this has been studied mostly using animal models and more evidence is needed to correlate changes in microbiota compositions with health markers. To provide further information regarding the impact of bariatric surgery treatments on the gut microbiota composition, in this work we compared the changes in clinical parameters and microbiota composition in subjects undergoing medical dietary treatment (MT), sleeve gastrectomy (SG) or Roux-en-Y gastric bypass (RYGB).

## MATERIAL AND METHODS

### Patient inclusion and clinical parameters

This study was conducted in accordance with the Declaration of Helsinki and approved by the Ethics Committee of the Faculty of Medicine, Pontificia Universidad Catolica de Chile. All study participants provided written informed consent. Participants of this study were recruited from candidates of the Obesity Program from Red de Salud UC-Christus. Eligible subjects were men or women, 18–60 years old with a body mass index (BMI) 30–50 kg/m$^2$. Women who were pregnant or with the intention to get pregnant were excluded. Other exclusion criteria include chronic antibiotic use, record of small intestine

and/or colon resection, intestinal inflammatory diseases and probiotic consumption. A total of 19 patients were recruited. Nine patients following medical dietary treatment (MT), based on a hypocaloric diet combined with moderate physical activity three times per week, in addition to a monthly doctor visit for 12 months. In addition, five recruited patients underwent Roux-en-Y gastric bypass (RYGB) and five sleeve gastrectomy (SG) with a 5-trocar technique as described previously (*Escalona et al., 2007*; *Boza et al., 2012*). Patients undergoing either RYBG or SG received nutrient supplementation during the follow-up period such as multivitamin supplements, iron, vitamin B12 and calcium, and were instructed to follow a hypocaloric diet.

Anthropometric and clinical parameters were obtained in a clinical assessment. The patients were evaluated at three opportunities (baseline evaluation before medical intervention, 6 and 12 months post treatment), which consisted of anthropometry measurements (weight and size), laboratory studies taking blood sample (lipids, HOMA and HbA1c). Patients were instructed to bring a homogenized fecal sample in a sterile container the day of the evaluation, which was transported in ice and immediately stored at $-80\,^{\circ}$C.

Descriptive statistical parameters (mean and standard deviation), and significant differences between clinical data were estimated by using a non-parametric unpaired Wilcoxon rank-sum test at 0.05 significance level. Mean differences between phylum changes were contrasted using a parametric unpaired $t$-test at 0.05 significance level. Pairwise Spearman Rank correlations between clinical parameters changes and bacterial variation were done using R statistical environment (*R Core Team, 2013*).

## DNA isolation

Fecal samples were thawed and 150 mg were used for total DNA isolation using the ZR Fecal DNA MiniPrep kit (Zymo Research, USA) following manufacturer instructions and using a Disruptor Genie device (Scientific Industries, USA). Total DNA concentration was measured in a NanoDrop 2000c device (Thermo Fisher Scientific, USA).

## Analysis of gut microbiota by 16S rRNA gene sequencing

Fecal DNA samples were diluted to 20 ng/µl in Nuclease-free water (IDT, USA) and submitted for Illumina MiSeq sequencing to Molecular Research DNA sequence services (MR-DNA, USA). The 16S rRNA gene V3–V4 variable region was amplified using the 341F and 785R primers (*Klindworth et al., 2013*), adding a barcode on the forward primer. The reaction was performed in 30 cycles using the HotStarTaq Plus Master Mix Kit (Qiagen, USA). After amplification, PCR products were checked in a 2% agarose gel. Multiple samples were pooled together and purified using calibrated Ampure XP beads (Agencourt Bioscience Corporation, USA). The pooled and purified combined PCR products were used to prepare a DNA library using TruSeq DNA LT Sample Prep Kit (Illumina, USA) following manufacturer instructions. Sequencing was performed using MiSeq platform (Illumina, USA) by paired-end sequencing.

The Quantitative Insights Into Microbial Ecology (QIIME, v1.9.1) software was used to analyze the 16S rRNA sequences (*Caporaso et al., 2010*; *Navas-Molina et al., 2013*). Briefly, chimera sequences were removed, then paired sequences joined and barcode was depleted.

Operational taxonomic units (OTUs) were picked by closed reference command and defined by clustering at 1% divergence (99% similarity) using as reference the GreenGenes database (*DeSantis et al., 2006*; *McDonald et al., 2012*) release 05-2013. Low sequence counts were filtered from BIOM table using the minimum value of count/sample between all samples. To compare phylum level changes, OTUs belonging to phyla with less than 1% representation were removed. Alpha and Beta diversity were calculated using QIIME. BIOM OTU table and *weighted* Unifrac tables were exported from QIIME to R environment (*R Core Team, 2013*) for statistical analysis and figure representation. The raw data reads obtained from the MiSeq platform were stored in the SRA NCBI online public database with accession number SRP076859 (http://www.ncbi.nlm.nih.gov/sra/SRP076859).

## qPCR amplification

In order to validate 16S rRNA gene sequencing results, quantitative PCR analysis was performed in fecal DNA samples using specific primers (Table S1) that were previously described (*Rinttilä et al., 2004*; *Fierer & Jackson, 2005*; *Frank et al., 2007*; *Bacchetti De Gregoris et al., 2011*). Amplification and detection were carried out in a StepOnePlus equipment (Applied Biosystems, USA), using 96-well optical plates MicroAmp Fast Optical (ThermoFisher, USA), filled with a mixture containing for each well 5 µl of PowerUp SYBR Master Mix or Fast SYBR Green Master Mix (Applied Biosystems, USA), 0.3 µM of each primer (0.3 µl each), 4.4 µl nuclease-free water (IDT, USA) and 1 µl of DNA previously diluted to 10 ng/µl. DNA samples were amplified with an initial hold of 50 °C for 2 min and a polymerase activation step of 95 °C for 2 min for PowerUp SYBR Master Mix, or 95 °C for 20 s for Fast SYBR Green Master Mix, followed by 40 cycles of a denaturation at 95 °C for 3 s and 62 °C for 30 s annealing and elongation. To verify a single amplification peak, a melting curve was performed by incrementing the temperature from 62 °C to 95 °C. All the samples were amplified in triplicate, and to correct primer efficiency, each plate contained a standard curve with ten-fold dilutions of genomic DNA of one species of the corresponding phylum, starting from 10 ng of DNA of the following microorganisms: *Firmicutes*: *Lactobacillus acidophilus* ATCC 4356; *Bacteroides*: *B. dorei* CL03T12C01, HM-718; NIAID, NIH; *Actinobacteria*: *Bifidobacterium longum* subsp. *infantis* ATCC 15697; *Proteobacteria*: *Escherichia coli* K12. The 16S rRNA gene copy number for each phylum in each sample was estimated from the corresponding standard curve and adjusted by the average genome 16S rRNA gene copy number of bacteria. To convert bacterial DNA amounts into copy number, the following equation was applied:

$$
Copynumber\ 16SrRNA\ gene
$$
$$
= \frac{Avogadro\ No\left(\mathrm{mol}^{-1}\right) * DNA\ quantity\left(\mathrm{g}\right) * Genome\ 16S\ copy\ number}{Genome\left(\mathrm{pb}\right) * 660\left(\frac{\mathrm{g}}{\mathrm{mol}}\right)}.
$$

## RESULTS

### Effect of obesity treatments on clinical parameters

In this study we compared the impact of three treatments for obesity on the gut microbiota, which included 19 patients who have undergone SG or RYGB, or received MT. Clinical

data and fecal samples were collected before the medical intervention, and after six and twelve months post-treatment.

Six months after surgery, both RYGB and SG patients showed a marked weight loss, accompanied by significantly lower BMI values (28% and 29% decrease) and significant waist and hip perimeter reduction (Table 1 & Table S2). In contrast, patients on MT showed similar weight after six months of treatment with no major improvements in their anthropometric parameters (2.8% reduction in BMI).

Other specific signatures were found in the clinical markers of this cohort (Table 1). In RYGB and SG treatments, we observed a decrease in insulin levels and glycemia, however these levels maintained in a normal range. Importantly, surgery led to a significant improvement in insulin sensitivity (HOMA < 3) after treatment. No statistical significant changes were observed in lipid metabolism markers, nevertheless cholesterol and triglycerides tended to decrease in RYGB after surgery (Table S2). In contrast to the above observations, none of these clinical parameters improved in MT patients. The general clinical observations presented here appeared stable after 12 months of treatment (Table S2).

## Obesity treatments induce global microbiota changes

Next, we determined the composition of the gut microbiota of these patients before and six months after each treatment. The hypervariable regions V3–V4 of the 16S rRNA gene in each fecal DNA sample was amplified by PCR and the products were sequenced by MiSeq Illumina platform. We obtained over 2 million reads in 38 samples, which includes DNA from before and after 6 months of obesity treatment. Rarefaction curves of the number of OTUs at different sequencing depths were obtained for each DNA sample (Fig. 1A), and they indicated saturation near 25,000 sequences. To evaluate microbiota composition of each patient, beta diversity was calculated, which estimates the degree of similarity of each sample in terms of their microorganism composition. Weighted Unifrac metric indicated that microbiota composition of MT patients was similar between 0 and 6 months (Fig. 1B). In contrast, the microbiota of patients undergoing RYGB cluster together at the beginning of the study, but after treatment its composition was divergent (Fig. 1C). Conversely, microbiota composition in SG group was divergent before the medical intervention but after treatment the compositions of three patients clustered closely (Fig. 1D).

On average, more than 99% of sequences aligned to the phyla *Firmicutes, Bacteroidetes, Actinobacteria* and *Proteobacteria*, with the first two being dominant in these samples (Fig. 2A). Changes in the representation of each phylum in patients in each treatment were then expressed as the ratio of the relative abundance after the intervention compared to pre-treatment (Figs. 2B–2D). In MT patients, no major changes were observed in any of these bacterial groups after six months, consistent with their poor response to the treatment. In contrast, important changes were observed in RYGB and SG patients. In the RYGB group, both *Bacteroidetes* and *Proteobacteria* increased in abundance (Fig. 2D), while in SG patients we observed an increase in *Proteobacteria*, but a decrease in *Bacteroidetes* (Fig. 2C). *Firmicutes* abundance was mostly unaffected in these patients. These different changes in microbiota also caused an increase in the *Bacteroides/Firmicutes* ratio in SG patients (Fig. 2E), and conversely a strong decrease in the RYGB group, consistent with

Medina et al. (2017), PeerJ, DOI 10.7717/peerj.3443

**Table 1 Clinical and anthropometric parameters before (0 month) and after each obesity treatment (6 months).** Significant *p*-values (Mann–Whitney test) are denoted by bold numbers.

| Treatment (time) | MT (0 months) | MT (6 months) | *p*-value | RYGB (0 months) | RYGB (6 months) | *p*-value | SG (0 months) | SG (6 months) | *p*-value |
|---|---|---|---|---|---|---|---|---|---|
| | | | | **Anthropometric data** | | | | | |
| Weight (kg) | $102.3 \pm 23$ | $99.5 \pm 23.7$ | 0.479 | $100.1 \pm 11.6$ | $72.1 \pm 11.2$ | **0.008** | $88.9 \pm 7.5$ | $62.7 \pm 4.2$ | **0.008** |
| BMI (kg/m$^2$) | $38.9 \pm 5.8$ | $37.8 \pm 6.7$ | 0.86 | $37.1 \pm 2.8$ | $26.7 \pm 3.1$ | **0.012** | $35.2 \pm 2.4$ | $24.9 \pm 2.9$ | **0.012** |
| Waist circumference (cm) | $107.8 \pm 13.4$ | $99.3 \pm 16.6$ | 0.269 | $102.6 \pm 12.5$ | $76.4 \pm 7.1$ | **0.021** | $95.2 \pm 10.5$ | $73.6 \pm 8$ | **0.015** |
| Hip perimeter (cm) | $116.0 \pm 10.4$ | $111.2 \pm 14.8$ | 0.374 | $116.4 \pm 7.8$ | $97.2 \pm 5.4$ | **0.016** | $105.3 \pm 4.5$ | $91.0 \pm 4.1$ | **0.036** |
| | | | | **Lipid metabolism** | | | | | |
| Cholesterol (mg/dL) | $185.6 \pm 35.4$ | $178.0 \pm 15.5$ | 0.825 | $216.8 \pm 88.5$ | $148.6 \pm 39.8$ | 0.095 | $174.2 \pm 21.1$ | $171.4 \pm 15.4$ | 0.841 |
| Triglycerides (mg/dL) | $109.0 \pm 28.8$ | $109.4 \pm 30.8$ | 0.965 | $349.0 \pm 443.5$ | $119.6 \pm 69.9$ | 0.151 | $151.6 \pm 87.3$ | $205.6 \pm 246.3$ | 0.841 |
| HDL (mg/dL) | $52.0 \pm 12.9$ | $56.1 \pm 122$ | 0.401 | $43.4 \pm 6.4$ | $46.6 \pm 10.6$ | 0.753 | $44.4 \pm 13.5$ | $50.0 \pm 17$ | 0.599 |
| LDL (mg/dL) | $111.8 \pm 32.2$ | $100.1 \pm 14.8$ | 0.48 | $103.8 \pm 19$ | $78.0 \pm 18$ | 0.095 | $99.0 \pm 15.8$ | $80.2 \pm 36.2$ | 0.6 |
| | | | | **Glucose metabolism** | | | | | |
| Glycemia (mg/dL) | $95.6 \pm 11.7$ | $90.4 \pm 13.2$ | 0.426 | $93.4 \pm 14.2$ | $78.2 \pm 4.4$ | **0.032** | $89.6 \pm 19.7$ | $75.4 \pm 2.4$ | 0.343 |
| Insulin (mcU/dL) | $39.6 \pm 25.4$ | $26.1 \pm 16.2$ | 0.136 | $16.7 \pm 7.9$ | $5.9 \pm 3$ | **0.008** | $14.3 \pm 5.3$ | $6.4 \pm 2.8$ | **0.036** |
| HOMA | $9.4 \pm 6.2$ | $6.1 \pm 4.4$ | 0.102 | $3.4 \pm 1.6$ | $1.2 \pm 0.7$ | **0.008** | $3.0 \pm 1.4$ | $1.2 \pm 0.5$ | **0.032** |
| HbA1c (%) | $5.9 \pm 0.5$ | $5.8 \pm 0.5$ | 0.894 | $5.5 \pm 0.1$ | $5.5 \pm 0.2$ | 0.589 | $5.5 \pm 0.5$ | $5.6 \pm 0.5$ | 0.916 |
| | | | | **Liver function** | | | | | |
| SGOT (UI/L) | $35.1 \pm 18$ | $31.3 \pm 19.5$ | 0.659 | $18.6 \pm 5$ | $16.4 \pm 1.9$ | 0.67 | $29.0 \pm 21.7$ | $15.6 \pm 3$ | 0.205 |
| SGPT (UI/L) | $57.2 \pm 34.9$ | $37.2 \pm 23.7$ | 0.216 | $17.6 \pm 5.9$ | $13.2 \pm 3.6$ | 0.243 | $28.6 \pm 14.4$ | $15.0 \pm 4.8$ | 0.206 |
| GGT (UI/L) | $48.2 \pm 40$ | $36.4 \pm 39$ | 0.17 | $17.0 \pm 1.9$ | $8.6 \pm 1.7$ | **0.011** | $23.2 \pm 14.1$ | $11.0 \pm 1.6$ | **0.016** |
| Phosphatase A (UI/L) | $82.8 \pm 16.8$ | $78.1 \pm 18.9$ | 0.895 | $55.0 \pm 34.5$ | $86.0 \pm 22.7$ | 0.222 | $85.2 \pm 22.1$ | $78.0 \pm 10.5$ | 0.548 |
| Total Bilirrubin (mg/dL) | $0.4 \pm 0.2$ | $0.5 \pm 0.2$ | 0.387 | $0.3 \pm 0.1$ | $0.4 \pm 0.2$ | 0.548 | $14.6 \pm 31.5$ | $0.4 \pm 0.3$ | 0.548 |
| Direct Bilirrubin (mg/dL) | $0.2 \pm 0.06$ | $0.2 \pm 0.1$ | 0.142 | $0.1 \pm 0.03$ | $0.2 \pm 0.1$ | 0.074 | $0.2 \pm 0.1$ | $0.2 \pm 0.1$ | 0.675 |
| Total Protein (%) | $97.1 \pm 15$ | $104.8 \pm 14.4$ | 0.401 | $98.4 \pm 4.7$ | $91.4 \pm 11.6$ | 0.248 | $97.0 \pm 5.1$ | $101.2 \pm 11.2$ | 0.527 |
| | | | | **Plasmatic data** | | | | | |
| Hto | $43.0 \pm 3$ | $42.3 \pm 3.3$ | 0.596 | $39.6 \pm 3.9$ | $35.9 \pm 4.9$ | 0.222 | $41.0 \pm 2$ | $38.3 \pm 3.2$ | 0.222 |

**Notes.**

MT, Medical Treatment; RYGB, Roux-en-Y Gastric Bypass; SG, Sleeve Gastrectomy.

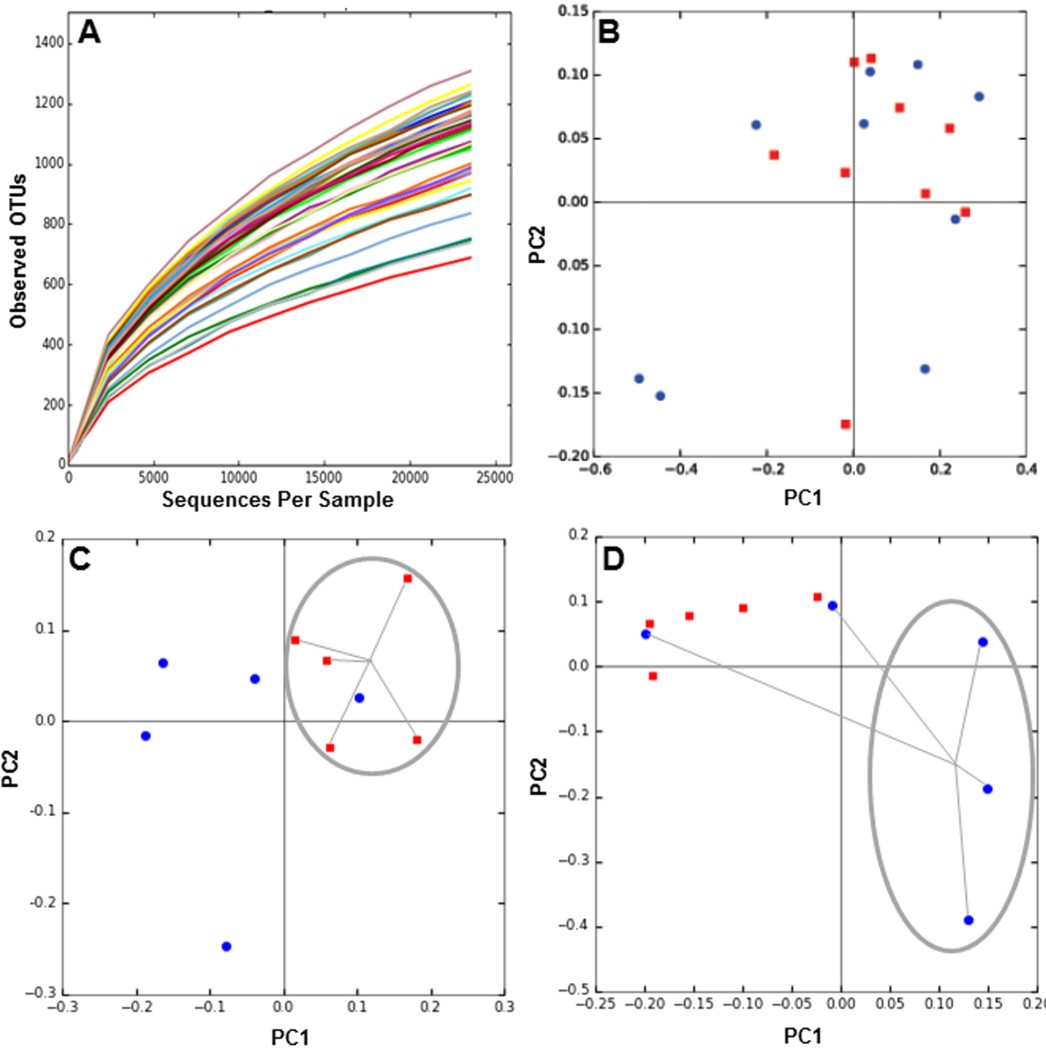

**Figure 1** **Rarefaction curves and Principal Component Analysis of the gut microbiota composition of the subjects of the study, obtained by 16S rRNA sequencing and QIIME.** (A) rarefaction curves for all samples sequenced, indicating the number of OTUs observed with different sequencing depths; (B–D) 2D-PCoA analysis of gut microbiota composition at time 0 (red squares) compared to 6 months after treatment (blue circles), for medical treatment subjects (MT), Sleeve Gastrectomy (SG) and Roux-en-Y Gastric Bypass (RYGB) groups.

previous reports (*Li et al., 2011*; *Kong et al., 2013*; *Walters, Xu & Knight, 2014*). Abundance and microbial composition obtained from 16S rRNA gene sequencing at genus-species level is summarized in Table S3 for each group.

In order to validate the 16S rRNA gene sequencing results, the composition of the microbiota of these patients was also evaluated by qPCR, quantifying the abundance of the phyla *Actinobacteria*, *Firmicutes* and *Bacteroides*, and the order *Enterobacteriales*. These results showed a good correlation with data obtained by MiSeq sequencing. MT patients did not show major microbiota changes after six months of treatment (Fig. 3A). In RYGB patients, we observed an increase in *Bacteroidetes* and a decrease

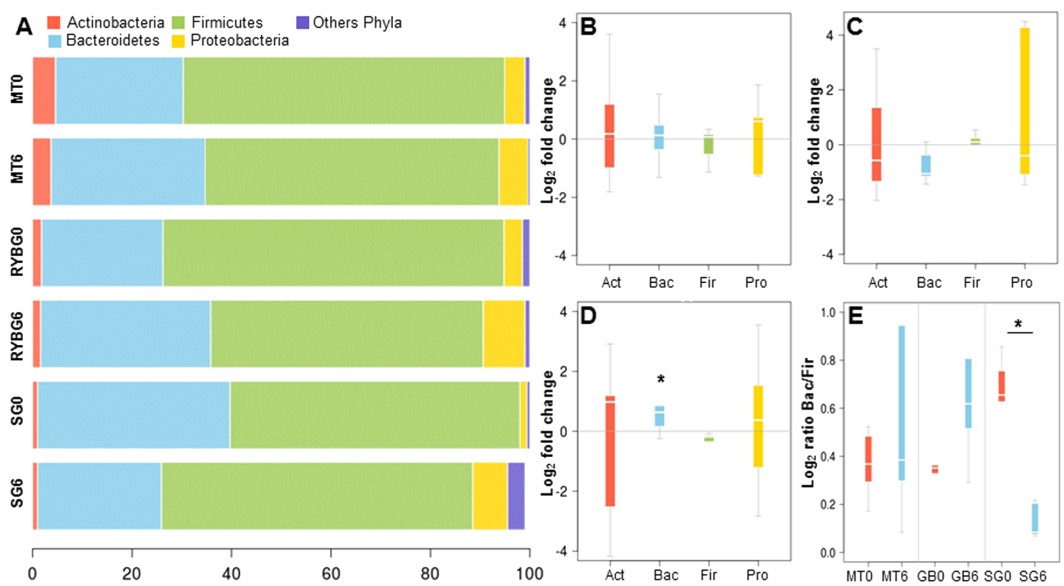

**Figure 2** **Relative abundance of major bacterial phyla in the gut microbiota of the three groups of the study, before and after each treatment.** (A) average relative abundance of four representative phyla (indicated in the upper part), at time 0 and 6 months, in medical treatment (MT), Sleeve Gastrectomy (SG) or Roux-en-Y Gastric Bypass (RYGB) groups; (B–D) changes in the abundance of each phylum after/before each treatment. The phyla are represented from left to right side as Act (*Actinobacteria*), Bac (*Bacteroidetes*), Fir (*Firmicutes*), Pro (*Proteobacteria*). Values were expressed as fold change and log(2) normalized; E: *Bacteroidetes/Firmicutes* ratio at time 0 (red) and 6 months after treatment (light blue) for Medical Treatment (MT), Roux-en-Y gastric bypass (GB) and Sleeve Gastrectomy (SG). Asterisk indicates significant fold change differences (*p*-value < 0,05).

in *Firmicutes* abundance (Fig. 3B). *Actinobacteria* also appeared in higher amounts in these patients. In contrast, in SG patients the abundance of *Bacteroidetes* decreased, while the *Firmicutes* and *Enterobacteriales* proportions increased (Fig. 3C). This in turn caused a reduction in the ratio *Bacteroidetes/Firmicutes* (Fig. 3D), similar to our previous observations by 16S rRNA gene sequencing (Fig. 2D). qPCR raw data and gene copy number for each group are included in Table S4, while phylum relative abundance measured by qPCR and 16S rRNA sequencing is presented in Table S5.

A correlation analysis comparing the abundance of each microbiota across all patients is represented in Fig. 4. This analysis shows that in general the initial microbiota abundance of most patients was uniformly correlated (R Spearman > 7). However, after six months of each treatment, microbiota abundance of RYGB subjects was very heterogeneous, in contrast to MT and SG subjects which grouped according to their respective treatment (Fig. 4B).

## Microbial signatures for each obesity treatment

The most abundant genera or species that displayed the highest change in abundance after each obesity treatment (determined by 16S rRNA sequencing) is shown in Table 2. These bacteria mostly belong to the *Firmicutes* phylum. Changes in the microbiota of MT patients were less pronounced compared to the surgical treatments. In the RYGB group, *Succiniclastum* sp. displayed a significant increase, as well as a few *Bacteroides* and *Citrobacter* species (Table 2). A *Bulleidia* OTU displayed the highest fold change in the SG

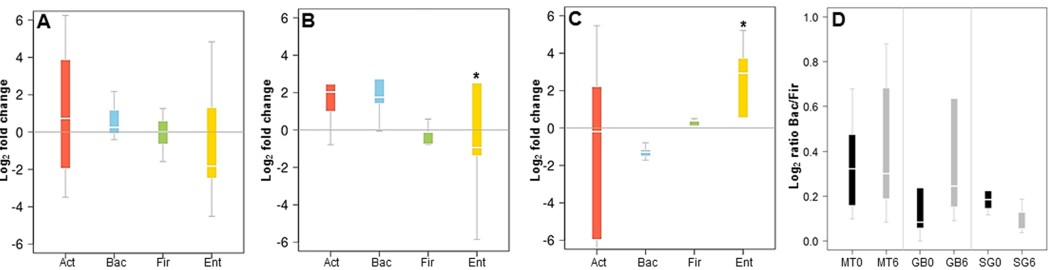

**Figure 3 Changes in major phyla in the gut microbiota after each obesity treatment measured by qPCR.** (A–C) ratio of the abundance of each phylum after/before each treatment for Medical Treatment (A), Roux-en-Y gastric bypass (B) and Sleeve Gastrectomy (C). The changes are represented from left to right side as Act (*Actinobacteria*), Bac (*Bacteroidetes*), Fir (*Firmicutes*), Ent (*Enterobacteriales*). Values were expressed as fold change and $\log_2$ normalized. (D) Ratio of *Bacteroidetes*/*Firmicutes* at time 0 (black) and six months after treatment (grey). Asterisk indicates significant fold change differences ($p$-value < 0.05).

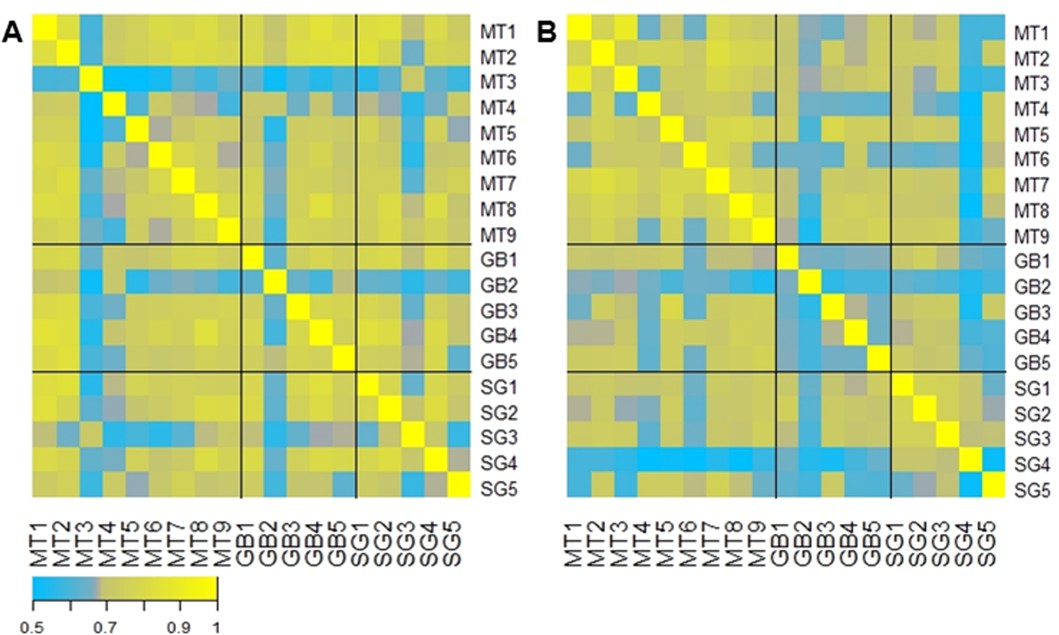

**Figure 4 Gut microbiota comparison across subjects before and after the obesity treatment.** Heatmap of pairwise Spearman Rank correlations for gut microbiota abundance, with other subjects in the study. (A) Pairwise correlations for all subjects before each intervention (Medical Treatment (MT), Roux-en-Y Gastric Bypass (GB) and Sleeve Gastrectomy (SG); numbers indicate subjects); (B) Pairwise correlations for all subjects six months after each treatment. The legend in the bottom indicates the scale of correlation across different gut microbiota compositions.

group, followed by *Escherichia coli* and *Akkermansia muciniphila* OTUs. Other interesting signatures observed include an increase of *Streptococcus luteciae* in both RYGB and SG patients. Species such as *Bacteroides eggerthii*, *Bacteroides coprophilus* and *Lactobacillales* sp. showed an important increase in RYGB subjects, but a marked down-representation after six months in the SG group.

Medina et al. (2017), *PeerJ*, DOI 10.7717/peerj.3443

**Table 2  Top ten OTUs (genus or species), with the highest fold change abundance between 0 and 6 months after each treatment.** Absolute abundance values were expressed as the average of each species abundance. Fold change is the $\log_2$ normalized ratio 6 to 0 months. Unpaired $t$-tests were used to find significant differences between averages.

| Medical treatment | 0 m | 6 m | Fold change | p-val | Roux-en-Y gastric bypass | 0 m | 6 m | Fold change | p-val | Sleeve gastrectomy | 0 m | 6 m | Fold change | p-val |
|---|---|---|---|---|---|---|---|---|---|---|---|---|---|---|
| Elusimicrobiaceae sp. | 0.11 | 4.89 | 5.46 | 0.34 | Lactobacillales sp. | 1.4 | 140.6 | 6.65 | 0.34 | Streptococcus luteciae | 9.6 | 1732.8 | 7.50 | 0.35 |
| RF32 sp. | 3.22 | 81.67 | 4.66 | 0.30 | Tenericutes sp. | 0.8 | 69.8 | 6.45 | 0.35 | Bulleidia p-1630-c5 | 2.2 | 199.8 | 6.50 | 0.34 |
| Peptococcaceae rc4-4 sp. | 0.22 | 2.44 | 3.46 | 0.20 | Succiniclasticum sp. | 1 | 86.2 | 6.43 | 0.35 | Streptococcus anginosus | 1.6 | 123.8 | 6.27 | 0.11 |
| Anaerobiospirillum sp. | 1.67 | 16.89 | 3.34 | 0.38 | Bacteroides coprophilus | 1.2 | 88.2 | 6.20 | 0.35 | Clostridium perfringens | 1.6 | 50.4 | 4.98 | 0.11 |
| Bacteroides ovatus | 344.11 | 1794.67 | 2.38 | 0.19 | Bacteroides eggerthii | 3.8 | 154 | 5.34 | 0.35 | Escherichia coli | 0.8 | 23.6 | 4.88 | 0.29 |
| Coprococcus catus | 0.11 | 0.56 | 2.32 | 0.18 | Mollicutes sp. | 1 | 33.6 | 5.07 | 0.32 | Leuconostocaceae sp. | 0.2 | 5 | 4.64 | 0.37 |
| Eubacterium cylindroides | 0.67 | 3.00 | 2.17 | 0.35 | Veillonella dispar | 3 | 90.2 | 4.91 | 0.18 | Gemellaceae sp. | 0.2 | 4 | 4.32 | 0.04 |
| Bifidobacterium bifidum | 51.33 | 206.22 | 2.01 | 0.27 | Coprobacillus sp. | 0.8 | 16.2 | 4.34 | 0.37 | Enterobacteriaceae sp. | 234.2 | 3921 | 4.07 | 0.33 |
| Lachnobacterium sp. | 57.44 | 218.56 | 1.93 | 0.42 | Clostridium bolteae | 16.8 | 287.8 | 4.10 | 0.35 | Mollicutes sp. | 0.4 | 6.4 | 4.00 | 0.17 |
| Atopobium sp. | 0.33 | 1.22 | 1.87 | 0.16 | Leuconostocaceae Leuconostoc sp. | 0.4 | 6.2 | 3.95 | 0.36 | Akkermansia muciniphila | 96.2 | 1351.8 | 3.81 | 0.19 |
| Akkermansia muciniphila | 119.44 | 11.67 | −3.36 | 0.10 | Coprococcus eutactus | 184.8 | 20.8 | −3.15 | 0.06 | Lactobacillales; sp. | 39.2 | 2.4 | −4.03 | 0.37 |
| Turicibacteraceae Turicibacter sp. | 19.11 | 1.78 | −3.43 | 0.13 | Peptococcaceae rc4-4 sp. | 2.2 | 0.2 | −3.46 | 0.34 | Dialister sp. | 2268.2 | 104.6 | −4.44 | 0.32 |
| Desulfovibrionaceae sp. | 3.78 | 0.22 | −4.09 | 0.23 | Lachnobacterium sp. | 351.8 | 26.4 | −3.74 | 0.36 | RF32; sp. | 174.8 | 7.4 | −4.56 | 0.36 |
| Leuconostocaceae sp. | 2.22 | 0.11 | −4.32 | 0.18 | Bifidobacterium animalis | 127.2 | 6 | −4.41 | 0.37 | Bifidobacterium bifidum | 65.6 | 2.4 | −4.77 | 0.29 |
| Serratia sp. | 5.22 | 0.22 | −4.55 | 0.30 | Bifidobacterium pseudolongum | 34.2 | 1.4 | −4.61 | 0.36 | Prevotellaceae Prevotella sp. | 710.2 | 22 | −5.01 | 0.36 |
| Lactobacillus sp. | 58.22 | 1.89 | −4.95 | 0.33 | Bulleidia p-1630-c5 | 429.8 | 12.2 | −5.14 | 0.05 | Paraprevotella sp. | 23.2 | 0.2 | −6.86 | 0.33 |
| Streptococcus luteciae | 264.78 | 7.11 | −5.22 | 0.32 | Lactobacillus ruminis | 355.8 | 9.2 | −5.27 | 0.26 | Lactobacillus sp. | 280 | 1.6 | −7.45 | 0.34 |
| Enterococcus sp. | 541.89 | 9.89 | −5.78 | 0.33 | Ruminococcus callidus | 242 | 4.4 | −5.78 | 0.09 | Bacteroides eggerthii | 501.6 | 2 | −7.97 | 0.35 |
| Clostridium baratii | 8.11 | 0.11 | −6.19 | 0.33 | Cyanobacteria YS2 sp. | 560 | 7.2 | −6.28 | 0.34 | Coprobacillus sp. | 116.4 | 0.4 | −8.18 | 0.34 |
| Enterococcus casseliflavus | 57.56 | 0.33 | −7.43 | 0.33 | Eubacterium cylindroides | 198.8 | 0.6 | −8.37 | 0.35 | Bacteroides coprophilus | 186.4 | 0.6 | −8.28 | 0.34 |

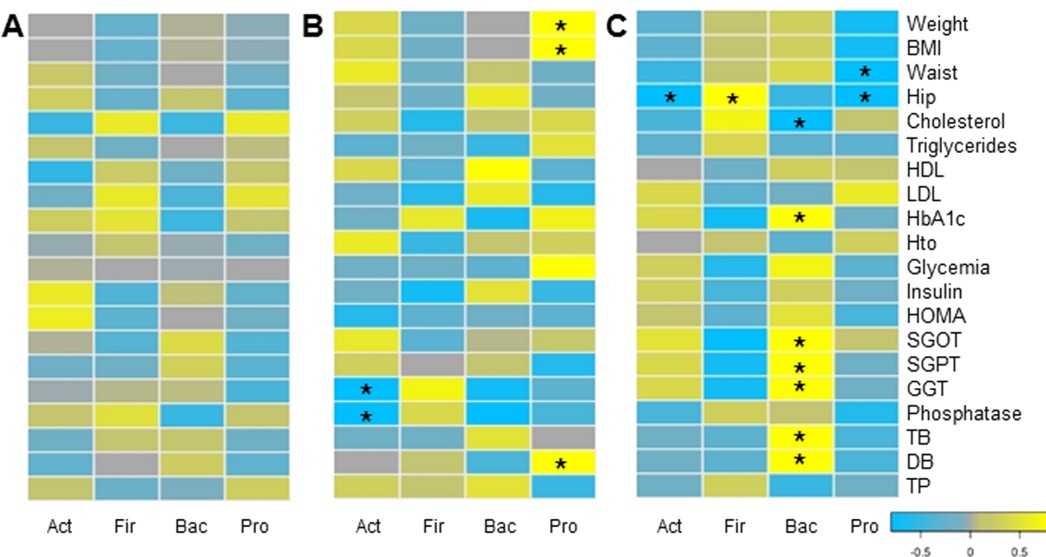

**Figure 5** **Correlations between clinical data change and gut microbiota variation.** Heatmap of pairwise Spearman Rank correlation analysis of microbiome changes with changes in clinical markers, considering data after and before the treatment. Data corresponds to subjects on Medical Treatment (A), Rouxen-Y gastric bypass (B) and Sleeve Gastrectomy (C). The phyla are represented from left to right side as Act (*Actinobacteria*), Fir (*Firmicutes*), Bac (*Bacteroidetes*) and Pro (*Proteobacteria*). Blue denotes a negative correlation while yellow indicates positive correlations. The asterisk denotes significant correlation values ($p$-value < 0.05).

## Associations between microbiota changes and clinical markers

Finally, we evaluated whether changes in microbiota composition in each group correlated with clinical and anthropometric data collected. Certain positive and negative associations between microbial changes and clinical markers after six months of each treatment were found, using Pairwise Spearman Rank correlation analysis (Fig. 5). Only a few of them were statistically significant ($p$-value < 0.05). In MT patients, no significant correlations were observed. Changes in *Proteobacteria* positively correlated with weight loss and bilirubin levels in RYGB patients. In addition, a negative association between *Actinobacteria* and liver markers such as Gamma-glutamyl transpeptidase and alkaline phosphatase was obtained. In the SG group, hip and waist perimeter reduction correlated negatively with *Proteobacteria* and *Actinobacteria,* and positively with *Firmicutes* abundance. In this group most significant clinical changes were associated with the reduction in *Bacteroidetes*, such as liver markers and HbA1c.

## DISCUSSION

In this study we compared the impact of three different obesity treatments on the gut microbiota and we investigated how these changes correlated with clinical markers. BS procedures are well known for causing a marked weight loss, and importantly reducing the incidence of comorbidities such as type 2 diabetes (*Sjöström et al., 2007*; *Sjöström, 2008*; *Eldar et al., 2011*). A direct contribution of the gut microbiota post-surgery in weight loss

and reduced adiposity has been shown in animal models (*Liou et al., 2013*). Moreover, the impact of these surgeries on the microbiota have been shown to be stable in the long-term (*Tremaroli et al., 2015*).

The size of the study was relatively small, which might limit certain observations. However, other similar studies have been conducted in small cohorts (*Zhang et al., 2009*; *Graessler et al., 2013*; *Damms-Machado et al., 2015*; *Tremaroli et al., 2015*; *Palleja et al., 2016*), and changes in the gut microbiota of these subjects are clear.

Both RYGB and SG induce different rearrangements in the gastrointestinal tract, and therefore it is expected that they also cause changes in the gut microbiota composition. Most studies to date have focused on the impact of RYGB procedures on the gut microbiota, and little research has been done on SG. Previous clinical studies have shown clear alterations in the microbiota associated to these treatments, especially in *Proteobacteria* (such as *E. coli*) (*Zhang et al., 2009*; *Furet et al., 2010*; *Kong et al., 2013*; *Tremaroli et al., 2015*; *Palleja et al., 2016*). Evidence in animal models indicate a similar pattern of microbiota alteration (*Liou et al., 2013*; *Shao et al., 2016*). In concordance with previous studies, an important increase in *Proteobacteria* in RYGB and SG was observed in this study. The overgrowth of these microbes could be associated with increases in luminal acidity and dissolved oxygen after these procedures, conditions that largely favour the growth of enterobacteria (*Duncan et al., 2009*). It could also be that *Escherichia* contributes to a more efficient energy harvest after BS during the initial nutritional starvation (*Tennant et al., 1968*).

In addition, we observed that the phylum *Bacteroidetes* was increased in RYGB patients, however it was down-represented in SG subjects. The physiological changes associated with these treatments, and their consequences may explain these differences, as *Bacteroides* species are dominant in the adult microbiome, and are well known for their foraging ability for complex polysaccharides. Furthermore, these species are also favoured by less acidic luminal pH (*Duncan et al., 2009*).

Previous studies regarding changes in microbiota associated to BS have also shown changes in key microorganisms of the microbiome. For example, *Akkermansia muciniphila* has a remarkable mucin degrading ability, and it has been shown to prevent inflammation and adipose tissue alterations (*Schneeberger et al., 2015*). Mouse and clinical data have shown an increase in the abundance of this species after RYGB (*Liou et al., 2013*; *Palleja et al., 2016*), and here we observed a similar trend on SG patients. *Faecalibacterium prausnitzii*, a key species in the gut microbiome (*Miquel et al., 2015*), appeared under-represented post RYGB (*Palleja et al., 2016*). Sequencing data in this study also revealed other signature species with interesting changes. *Streptococcus luteciae* (*Firmicutes*) increased its abundance means several fold in RYGB and SG, but decreased in MT. A *Lactobacillales* OTU, *Bacteroides coprophilus* and *Bacteroides eggerthii* were increased after RYGB but were underrepresented post SG surgery. Conversely, a *Bulleidia* OTU (*Firmicutes*) showed the highest increase after SG but reduced its abundance several fold post RYGB. It is possible that the enrichment or depletion of these microorganisms in the gut microbiota might contribute to the positive health outcomes of RYGB and SG.

It is important to take into consideration that fecal microbiota is more representative of the large intestine, and evaluating the changes in upper parts of the intestine is difficult to achieve (*Goodrich et al., 2014*). Moreover, gut microbiota composition is influenced by geography distribution (*De Filippo et al., 2010*; *Yatsunenko et al., 2012*). Dietary and caloric restriction might also have a direct impact in the gut microbiota after obesity surgery (*Buchwald et al., 2004*). However, in this study we observed that medical treatment, consisting of an intervention with a low calorie diet and physical activity, does not result in significant changes in the gut microbiota or weight markers.

Finally, we found certain significant associations between changes in bacterial phyla abundance and clinical parameters after surgery. For example, changes in *Proteobacteria* correlated negatively with weight and BMI in RYGB patients, and in SG patients we observed that *Bacteroidetes* correlated with certain blood and hepatic markers. It would be interesting to determine if these associations in humans are causative or are a consequence of each surgical procedure. Unfortunately, this study is correlational, and therefore further functional studies are needed to understand the role of the gut microbiota in weight loss and metabolic improvements observed after bariatric surgery, as observed in animals (*Liou et al., 2013*; *Tremaroli et al., 2015*). At least in mice, two studies have shown that the microbiota post-RYGB is capable of transmitting weight loss and reduced adiposity to germ-free mice, indicating a causal relationship of the microbiota influencing metabolic processes (*Liou et al., 2013*; *Tremaroli et al., 2015*). In humans, a recent report showed that fecal microbiota transplantation from a healthy but overweight donor induced obesity in one person (*Alang & Kelly, 2015*).

It is probable that advances in functional metagenomics, and measuring additional clinical markers such as bile salts and hormone production, could provide a better description of the role of the gut microbiota in health in the context of bariatric surgery. Also including larger cohorts, evaluated for longer periods, might provide more solid answers. These studies will definitively help to improve surgical procedures, and eventually design microbiome based bacto-therapies aimed to treat metabolic disorders.

## CONCLUSIONS

In this study we determined the impact of two bariatric procedures at the same time scale on the gut microbiota, including a comparison with the effect of medical treatment on human subjects. While no major changes were observed on weight, clinical markers or the microbiota of subjects on MT, both RYGB and SG caused major adjustments in the gut microbiota, which correlated with certain anthropometric or metabolic parameters. An increase in *Proteobacteria* was observed six months after both RYGB and SG, whereas *Bacteroidetes* increased in RYGB but decreased in SG. These alterations are probably caused by the physiological rearrangements of the gastrointestinal tract, and may in fact contribute to weight and metabolic improvement in these subjects.

### Funding

This work was supported by Proyecto Interdisciplina UC No 10/2013, Fondecyt Postdoctorado [3160525] and Fondecyt de Iniciacion [11130518]. The funders had no role in study design, data collection and analysis, decision to publish, or preparation of the manuscript.

### Grant Disclosures

The following grant information was disclosed by the authors:
Proyecto Interdisciplina UC: 10/2013.
Fondecyt Postdoctorado: 3160525.
Fondecyt de Iniciacion: 11130518.

### Competing Interests

The authors declare there are no competing interests.

### Author Contributions

- Daniel A. Medina conceived and designed the experiments, performed the experiments, analyzed the data, wrote the paper, prepared figures and/or tables, reviewed drafts of the paper.
- Juan P. Pedreros performed the experiments, analyzed the data, prepared figures and/or tables.
- Dannae Turiel performed the experiments, analyzed the data.
- Nicolas Quezada and Fernando Pimentel conceived and designed the experiments, contributed reagents/materials/analysis tools, reviewed drafts of the paper.
- Alex Escalona conceived and designed the experiments, contributed reagents/materials/analysis tools, wrote the paper, reviewed drafts of the paper.
- Daniel Garrido conceived and designed the experiments, analyzed the data, contributed reagents/materials/analysis tools, wrote the paper, prepared figures and/or tables, reviewed drafts of the paper.

### Human Ethics

The following information was supplied relating to ethical approvals (i.e., approving body and any reference numbers):

This study was conducted in accordance with the Declaration of Helsinki and approved by the Ethics Committee of the Faculty of Medicine, Pontificia Universidad Catolica de Chile (CEI-MEDUC 13-288).

### Data Availability

The raw data reads obtained from the MiSeq platform were stored in SRA NCBI online public database with accession number SRP076859 (http://www.ncbi.nlm.nih.gov/sra/SRP076859). Clinical data has been provided as Table S2.

## Supplemental Information

Supplemental information for this article can be found online at http://dx.doi.org/10.7717/peerj.3443#supplemental-information.

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
