# Peer review of "Distinct patterns in the gut microbiota after surgical or medical therapy in obese patients"

_PeerJ, doi:10.7717/peerj.3443_

## Round 0.1 · original submission · Major Revisions

This study reported the distinct patterns in gut microbiota in obese patients with medical treatment, sleeve gastrectomy and Roux-en-Y gastric bypass. In general, the study was well-designed and nicely conducted, and the results obtained showed interesting changes in gut microbiota, which are correlated with clinical observations. Reviewers found these findings are important and the manuscript is well-written. Listed below are additional comments:

1. The statistical analysis for qPCR and Figure 5 should be described, and the statistical analysis for Table 2 would be desired.
2. Literature citation, such as Huttenhower et al., 2012; Caporaso et al., 2010 had more than 10 authors. Some journals limit 3 authors, some 6 authors, and I would suggest to use “et al.” after 10 authors.
3. The qPCR raw data would be better included as supplemental data. As each bacteria category have thousands to million entries. Some primers worked, while some did not. What were the Ct values in your qPCR analysis?

Reviewer 1 ·

Basic reporting

see below

Experimental design

see below

Validity of the findings

see below

Additional comments

The authors compared the impact of medical treatment, sleeve gastrectomy and Roux-en-Y gastric bypass on the gut microbiota from obese subjects. The authors revealed interesting changes in two phyla, namely Proteobacteria and Bacteroidetes, which are positively correlated with changes in weight and BMI, or glucose metabolism. This is a very important finding, and the paper is well presented.

Reviewer 2 ·

Basic reporting

1. Cite more references for paragraph 1 in Introduction especially for diabetes
2. Please make sure all font size in figures are the same
3. Please highlight brief description in each figure legend
4. Put the panels in the right direction in Figure 2&3

Experimental design

1. Please describe advantages and disadvantages of current study in the Discussion part.
2. Please note future direction in the Discussion part

Validity of the findings

1. Need to revise data layout in order to become more clear and straightforward

Additional comments

I would accept this manuscript based on the revisions mentioned above.

Reviewer 3 ·

Basic reporting

Overall, the article by Medina et al aims to characterize microbial changes among weight loss treatment (medical, gastric bypass, and sleeve gastrectomy) and find correlations to clinical readouts. There is little clinical data showcasing changes in microbial composition for sleeve gastrectomy, so this article adds to current understanding. The article is written within professional article structure. However, there are few areas which require further proofing of grammar and spelling. There are also a few sections mentioned below that require some clarification. Please see the following comments:
Introduction -
Line 65-66 "these acid modulate physiological processes in several tissues" -- such as? mention a couple to put into broader appreciative context
Line 89 "showed that changes in microbiota composition are stable across the time, sustaining..." -
Lines 91-93 "On the other hand, FMT is effective treatment for recurrent Cdif...However, recent case report..." - the use of "on the other hand" and "however" is confusing and lacks a clear connection between microbial changes with RYGB, C.diff, and FMT-induced obesity. The way this is written, it particularly makes the statement about C.diff seem out of place within the context of this paragraph. Please explain more clearly why this is included and emphasize how changes in microbial composition is related to weight loss.
Line 93 "recently" - change to 'recent'
Line 96 "could influence the weigth gain on transplanted patient" - check spelling and grammar.
Lines 97-99 "Sleeve gastrectomy and...to different health outcomes" - Although there is a paragraph dedicated to the anatomical differences between the surgeries, there is no mention of what the major differences in physiological changes and health outcomes are between the two surgeries. It would be nice to highlight this, as it is a main reason why you are interested in characterizing the microbiota between the surgeries.

Experimental design

Primary research is within aims and scope with statement indicating gap in knowledge (primarily of microbial changes with SG).

Please clarify a few methodological questions, which may help the reader to interpret the findings better:
- Line 116 What is the monthly 'medical control' ?
- Line 122 - please clarify whether time 0 is before surgery or directly after surgery.
Given the variability evidence by Figures 1 and 4, this study may not be adequately powered (total 19 patients - 9 medical, 5 RYGB, 5 SG). Was there a pre-study statistical power analysis to confirm these numbers are adequately powered? If not, I would recommend mentioning this as a potential limitation in your Discussion.

Validity of the findings

Please see the following comments/suggestions below:

Table 1. The text mentions data for 0, 6, and 12 months post-treatment , but the Table only has up to 6 months (line 186). Please correct text.
Line 194 - text mentions RYGB causes significant decrease in cholesterol and LDL, but P value (0.095) does not indicate significance. Change to trend.
Line 195 - "Circulatinginsulin" - check spacing.
Line 196 - "glycemia was significantly lower in RYGB" --> Based on HbA1c and normal glucose levels, this patient population is not diabetic. Highlighting a reduction from normal glucose levels to lower normal glucose levels does not seem clinically relevant. The increased insulin sensitivity, however, is as expected and should be emphasized more.
Fig 1 Line 209-211. Please clarify what this sentence means. Are you saying RYGB and SG microbiomes are different from the start (suggesting time 0 is after surgery?). Any explanation for the 2/5 outliers in the SG 6 month treatment, since it appears that those two samples did not change from day 0.
Fig 2A Taxonomic level - how variable are the individual microbiomes before and after treatment? A bit concerning that the 0 time points for all patients is a wide range that also encompasses the 6 month treatment (e.g. SG0 Bacteroidetes is higher than RYGB0 or MT0), making it difficult to compare.
Figure 2 - mis-cited figures in text. Line 216: Figure 2B should be Figs 2B-D. Line 220: Fig 2B should be Figure 2D. Line 223 Figure 2D should be Figure 2E
Is there a reason for limiting the 16S RNA Sequencing Data to Phylum level? Would it be possible to obtain a bit more resolution in the microbial composition profiles in addition to phylum and high abundant OTUs?
Figure 3. Line 228. The text mentions qPCR was used to quantify abundance of Enterobacteriales, but there is no data in the figures to show this.
Figure 4. Can you explain the outliers in these heat maps (e.g. MT3 , GB2, and SG3 at time 0; GB2, SG4 at time 6)? It seems like there's a lot of patient-to-patient variability.
Figure 5. Please add more clarity in the measures taken to correlate clinical markers to microbial changes. There appears to be a disconnect between significant changes in clinical measures in Table 1 and the significant correlation heatmap on Fig 5.

Conclusions - Overall, it is clearly stated that there are differences in microbial compositions between surgery vs. medical treatment and that these changes may be correlated to clinical markers. However, what is not clear is whether there is meaningful differences between RYGB and SG. The clinical markers seem relatively similar (and those that are different are relatively small changes) despite some differing microbial findings. Would looking at other clinical markers or a follow-up study with a more severe diabetic cohort help further support host-microbe changes?

Additional comments

Overall, this is a complete unit of work that would add to the scientific database. There is a need to clean up some grammatical and spelling errors and also clarify sentence structures for introduction. I would also recommend more resolution in the data analysis and more clarity in the measures taken to correlate clinical markers to microbial changes.

Reviewer 4 ·

Basic reporting

The introduction needs background about obese microbiota profiles compared to lean - need a better picture of baseline state so we can contextualise the differences seen after surgery. 

Minor comments:

- Need to state the study design: this could also benefit from a flow chart with clear display of different treatment conditions/ durations

- line 114 - ‘up to 19 patients were recruited’, needs to be specific, was it 19 or was it less? 
- line 119 - did all patients receive this nutrition supplementation? Differences in dietary intake will be important to describe, as these relate directly to microbiota composition and may confound the analysis. 
-  line 154 - ‘Alfa’, should be alpha
-  line 159 Please include a statement about the purpose of the QPCR analysis, ie. ‘in order to … we performed QPCR’.

Experimental design

The research question is well defined and the topic of substantial interest. However, there are fundamental limitations that limit the usefulness of these data. These are discussed in the next section.

I do not have extensive expertise that would allow me to identify problems with the methodologies used in this study, but from my basic knowledge of the emerging field of microbiota research, the methodologies used to do the gene sequencing and to interpret the data are sound.

Validity of the findings

The fundamental problem with this study is the small sample size. Based on the newest insights in this emerging field (Sze MA, Schloss PD. 2016. Looking for a signal in the noise: revisiting obesity and the microbiome. mBio 7(4):e01018-16. doi:10.1128/mBio.01018-16), this study was only powered to detect an effect size of 15% or more.

The second issue is that there were no data on dietary intakes collected or analysed. The authors need to rule out diet as a confounder in their analysis. i.e. show that there are no differences between diets of treatment/ ctl groups. The microbiota would vary as a function of diet, and the type of surgery might be interacting with dietary intakes post-surgery. For example, one type of surgery may make the consumption of high-fibre foods more problematic, which would then influence dietary habits post-surgery.

Additional comments

The introduction needs background about obese microbiota profiles compared to lean - need a better picture of baseline state so we can contextualise the differences seen after surgery. 
Minor comments:
- Need to state the study design: this could also benefit from a flow chart with clear display of different treatment conditions/ duration
- line 114 - ‘up to 19 patients were recruited’, needs to be specific, was it 19 or was it less? 
- line 119 - did all patients receive this nutrition supplementation? Differences in dietary intake will be important to describe, as these relate directly to microbiota composition and may confound the analysis. 
-  line 154 - ‘Alfa’, should be alpha
-  line 159 Please include a statement about the purpose of the QPCR analysis, ie. ‘in order to … we performed QPCR’.The research question is well defined and the topic of substantial interest. However, there are fundamental limitations that limit the usefulness of these data. These are discussed in the next section.

I do not have extensive expertise that would allow me to identify problems with the methodologies used in this study, but from my basic knowledge of the emerging field of microbiota research, the methodologies used to do the gene sequencing and to interpret the data are sound.

The fundamental problem with this study is the small sample size. Based on the newest insights in this emerging field (Sze MA, Schloss PD. 2016. Looking for a signal in the noise: revisiting obesity and the microbiome. mBio 7(4):e01018-16. doi:10.1128/mBio.01018-16), this study was only powered to detect an effect size of 15% or more.

The second issue is that there were no data on dietary intakes collected or analysed. The authors need to rule out diet as a confounder in their analysis. i.e. show that there are no differences between diets of treatment/ ctl groups. The microbiota would vary as a function of diet, and the type of surgery might be interacting with dietary intakes post-surgery. For example, one type of surgery may make the consumption of high-fibre foods more problematic, which would then influence dietary habits post-surgery.

I believe this to be a topic of serious study. I would urge the authors to continue their investigations, but include an assessment of dietary intake and work to increase the samples size before submitting for publication. If there were the possibility of collecting data from a fourth group - those who have had lapband surgery - this would very much strengthen the study. Lapband surgery supports dramatic weight loss without the structural changes to the stomach that are likely to have a direct impact on microbiota.

---

## Round 0.2 · accepted · Accept

Please note that the reviewer#3 raised some minor comments "Figure 3 --> It would be helpful to also include which treatment are for A-D within the figure legend. Accordingly, in the text (line 249), placement of "(Figure 3C)" is missing", which should be done during the galley proof.

Other reviewers are satisfied with the reversion. Thanks for your
contribution.

Reviewer 3 ·

Basic reporting

The revised manuscript is much clearer throughout with better introduction and outline of study design.

Please consider the following minor adjustment:
Figure 3 --> It would be helpful to also include which treatment are for A-D within the figure legend. Accordingly, in the text (line 249), placement of "(Figure 3C)" is missing.

Experimental design

Changes made are much appreciated.

Validity of the findings

no comment.